# Chemical Source Searching by Controlling a Wheeled Mobile Robot to Follow an Online Planned Route in Outdoor Field Environments [note 1]

**DOI:** 10.3390/s19020426

**Published:** 2019-01-21

**Authors:** Ji-Gong Li, Meng-Li Cao, Qing-Hao Meng

**Affiliations:** 1Tianjin Key Laboratory of Information Sensing and Intelligent Control, School of Automation and Electrical Engineering, Tianjin University of Technology and Education, Tianjin 300222, China; charles75@163.com; 2Institute of Robotics and Autonomous Systems, Tianjin Key Laboratory of Process Measurement and Control, School of Electrical and Information Engineering, Tianjin University, Tianjin 300072, China; menglicao@tju.edu.cn; 3Logistics Engineering College, Shanghai Marinetime University, Shanghai 201306, China

**Keywords:** wheeled mobile robot, chemical-source tracing, outdoor field environments, chemical-patch path (C-PP), search-route planning

## Abstract

In this paper, we present an estimation-based route planning (ERP) method for chemical source searching using a wheeled mobile robot and validate its effectiveness with outdoor field experiments. The ERP method plans a dynamic route for the robot to follow to search for a chemical source according to time-varying wind and an estimated chemical-patch path (C-PP), where C-PP is the historical trajectory of a chemical patch detected by the robot, and normally different from the chemical plume formed by the spatial distribution of all chemical patches previously released from the source. Owing to the limitations of normal gas sensors and actuation capability of ground mobile robots, it is quite hard for a single robot to directly trace the intermittent and rapidly swinging chemical plume resulting from the frequent and random changes of wind speed and direction in outdoor field environments. In these circumstances, tracking the C-PP originating from the chemical source back could help the robot approach the source. The proposed ERP method was tested in two different outdoor fields using a wheeled mobile robot. Experimental results indicate that the robot adapts to the time-varying airflow condition, arriving at the chemical source with an average success rate and approaching effectiveness of about 90% and 0.4~0.6, respectively.

## 1. Introduction

Odor information is widely used by animals when searching for food, finding mates, exchanging information, and evading predators. Inspired by these olfactory search activities, in the early 1990s, researchers started trying to build mobile robots with similar olfaction abilities (or chemical sensing capabilities in general) to replace trained animals [1,2,3,4]. It is expected that mobile robots developed with such olfaction capabilities will play an increasing role in areas such as thwarting terrorist attacks, finding toxic or harmful gas-leakage locations, checking for contraband (e.g., heroin), and searching for survivors in collapsed buildings or in the water [5,6,7]. 

Research into the use of one or more mobile robots equipped with odor/gas sensors (in some research also with wind sensors) to search for chemical source(s) was called odor source localization (OSL) [2,8]. To make the OSL research more general, here we call it chemical source searching (CSS for short). The research into CSS can be roughly classified into behavior-based methods and analytical-model-based methods [7], with most works focusing on behavior-based methods. A behavior-based CSS procedure can commonly be decomposed into three subprocedures (namely, plume finding, plume traversal, and source declaration), according to Hayes et al. [8]. During the plume finding subprocedure, the robot tries to make contact with the chemical plume. Only a few publications have discussed plume-finding methods. Hayes et al. [8] proposed an initial outward spiral search pattern for plume finding. Recently, Marjovi et al. [9,10] discussed the optimal spatial configuration of swarm robotic gas sensors for plume finding. Once the plume is detected, the robot will switch to the second subprocedure, and begins tracing the chemical toward its source. Most methods for this subprocedure are biologically inspired, such as the gradient-following algorithms [11,12,13,14], the zigzagging algorithms [2,13], the upwind algorithms [8,15], adapted ant colony optimization algorithm [16], particle swarm optimization algorithm and its variants [17,18,19,20], the SPIRAL algorithm [21], and genetic programming algorithm [22]. During the plume tracing procedure, how to re-contact a lost plume is also an important issue [23]. In the final phase, the robot locates the source [24,25].

Conventional plume tracing methods, especially biologically inspired methods, make it quite hard for a wheeled mobile robot equipped with normal gas sensors to approach a previously unknown chemical source in outdoor airflow environments. This is because animals fundamentally differ in their sensing and actuation capabilities from state-of-the-art gas-sensitive mobile robots [26]. The maximum driving and steering speeds of normal wheeled mobile robots are limited by many factors, such as kinematical constraints, ground conditions, and considerations for purposes of safety. Thus, outdoor plumes usually change much faster than the speed of ground mobile robots. For example, according to our measurements in real outdoor environments, the airflow speed is usually significant (typically > 0.2 m/s, always above 1 m/s in windy days), and the airflow direction often changes randomly, rapidly, and substantially. It is well known that the chemical dispersal in airflow environments depends mainly on the transportation of turbulent airflow, forming an intermittent, patchy, sparse, and even meandering plume. Furthermore, the response/recovery lag and relatively low sensitivity of widely used metal-oxide-semiconductor gas sensors make the robot trace fast-changing and sparse plume harder. 

The main contribution of this paper is that an estimation-based route planning (ERP) method is proposed for controlling a single ground mobile robot to search for a chemical source according to the historical trajectory by which a chemical patch detected by the robot has passed, namely the chemical-patch path (C-PP), in the second subprocedure of CSS. Experimental results show that the ERP method allowed the robot to successfully track intermittent, sparse and random chemical plumes and finally approach the previously unknown source in two different outdoor environments of thousands of square meters. To our knowledge, most existing CSS methods were designed for indoor airflow environments, with little research involving real-robot platforms in outdoor field environments, and there is no report of successfully solving the whole CSS problem using a single wheeled mobile robot in thousands-of-square-meter outdoor environments. The ERP method not only takes into account the fact that the airflow direction can change radically in outdoor natural-airflow environments, but also has low requirements on gas sensors and the speed of robots. Thus, it can solve the primal problem of plume sparsity and fluctuation in plume traversal subprocedure, which impose great difficulties on conventional plume tracing methods.

The proposed ERP method consists of two steps. Firstly, the C-PP is estimated using historical airflow speeds and directions when a chemical patch is detected by the robot. Secondly, a search route is planned for the robot based on the estimated C-PP. In fact, the concerned chemical patch originally comes from the source. Therefore, the source should be covered by the estimated C-PP if the C-PP is correctly reckoned back enough in the time. The search route is designed by aiming to find more chemical patches and then re-calculate the C-PP, re-plan the search route, forming an iterative searching procedure, and finally making the robot arrive at the source. A rule of thumb is that it is easier to re-meet the chemical clue when the robot moves in the downwind region (than in the upwind region) of the chemical source. Thus, in our method, the search route is planned in the downwind region and close to the estimated C-PP. 

The remainder of this paper is organized as follows. The proposed CSS solution, which includes a C-PP estimation method, search-route planning method, and chemical-source searching procedure, is described in Section 2. In Section 3, the experimental platform is introduced. Section 4 presents the experiments I, which include the approximate uniformity test of wind field and determination of probability-density threshold. The experiments II, i.e., the chemical source searching based on online planned routes, are introduced in Section 5. Conclusions are given in Section 6.

## 2. The Proposed ERP Method

The proposed ERP solution consists of two main steps, C-PP estimation and search route planning, in which the search route is planned based on the estimated C-PP and time-varying wind. 

### 2.1. C-PP Estimation

#### 2.1.1. Probability Model of Chemical Transportation

Because a wheeled mobile robot mainly moves in 2D (two dimensional) ground environments to search for chemical sources, the wind field and gas dispersal are also assumed in a 2D horizontal plane. We construct a time sequence of the airflow, ${\left\{U(L_{R}(t_{i}))\right\}}_{i=0}^{k}$, in which $U(L_{R}(t_{i}))$ denotes the airflow velocity ${\left[u_{x}(L_{R}(t_{i})),u_{y}(L_{R}(t_{i}))\right]}^{T}$ (i.e., the velocity components in the *x* and *y* directions) observed by the robot at position $L_{R}(t_{i})$ at time $t_{i}$. Here, times $t_{0}$ and $t_{k}$ stand for the start time and current time, respectively. Over a short time scale, the movement of the chemical patch can be modeled as a random walk superimposed on a mean velocity [27,28,29]. Let $p_{*j}(t_{l},t_{j})$ stand for the probability density that a chemical patch detected by the robot at the position $L_{R}(t_{j})$ at time $t_{j}$ came from the position $L_{*}$ at time $t_{l}$
$\left(t_{j}>t_{l}\right)$. Then $p_{*j}(t_{l},t_{j})$ can be expressed [29] as
(1)$$p_{*j}(t_{l},t_{j})=\frac{1}{2\pi (t_{j}-t_{l})\sigma_{x}\sigma_{y}}e^{-\frac{{\left(\Delta x\left(t_{l},t_{j}\right)\right)}^{2}}{2\left(t_{j}-t_{l}\right)\sigma_{x}^{2}}}e^{-\frac{{\left(\Delta y\left(t_{l},t_{j}\right)\right)}^{2}}{2\left(t_{j}-t_{l}\right)\sigma_{y}^{2}}}$$
where ${\left[\sigma_{x}^{2},\sigma_{y}^{2}\right]}^{T}$, the variances of the airflow velocity, can be estimated online by the time sequence of the airflow, $\Delta x(t_{l},t_{j})=x_{j}-x_{*}-s_{x}(t_{l},t_{j})$, $\Delta y(t_{l},t_{j})=y_{j}-y_{*}-s_{y}(t_{l},t_{j})$, where $\left(x_{*},y_{*}\right)$ and $\left(x_{j},y_{j}\right)$ are the coordinates of the positions $L_{*}$ and $L_{R}(t_{j})$, respectively, and ${\left[s_{x}(t_{l},t_{j}),s_{y}(t_{l},t_{j})\right]}^{T}$ is the distance traveled by the air mass carrying the chemical patch concerned during the period from time $t_{l}$ to $t_{j}$.

With the assumption of approximately uniform airflow in an open field (see the test results presented in Section 4.1), the distance traveled by the air mass carrying the chemical patch concerned can be estimated as
(2)$${\left[s_{x}(t_{l},t_{j}),s_{y}(t_{l},t_{j})\right]}^{T}\approx \Delta t\sum_{i=l}^{j}U(L_{R}(t_{i}))$$
where Δt is the sampling period of the robot system (set to 0.5 s, in our experiments). Over a short time scale, only the 20 most recent elements in the time sequence of the airflow are used in this research, with a sampling period Δt=0.5 s. For these 20 most recent records, ${\left\{U(L_{R}(t_{i}))\right\}}_{i=f}^{j}$, the variance of the airflow velocity ${\left[\sigma_{x}^{2},\sigma_{y}^{2}\right]}^{T}$ in Equation (1) can be calculated online as
(3-a)$$\sigma_{x}^{2}=\mathrm{var}(u_{x}(L_{R}(t_{i}))),i=f,\cdots j$$
(3-b)$$\sigma_{y}^{2}=\mathrm{var}(u_{y}(L_{R}(t_{i}))),i=f,\cdots j$$
where f is the subscript for the time corresponding to the earliest element in the most recent 20 airflow records and can be calculated by
(4)f=max(0,j−20)

#### 2.1.2. C-PP Estimation Algorithm

When a chemical patch is detected by the mobile robot at $L_{R}(t_{j})$, its historical trajectory (i.e., the C-PP) can be estimated [30] as
(5)$$OW(t_{j})=\overset{j-1}{\underset{l=f}{\cup }}OS(t_{l},t_{j})$$
where $OS(t_{l},t_{j})$ is the area covering the possible locations of the concerned chemical patch at time $t_{l}$
(l≤j). Therefore, $OW(t_{j})$ represents all the areas through which the chemical patch possibly passed from time $t_{f}$ to time $t_{j-1}$. To cut down the calculation, $OS(t_{l},t_{j})$ can be estimated as
(6)$$\begin{array}{l} OS(t_{l},t_{j})=\{L_{*}\in W|p_{*j}(t_{l},t_{j})\geq \eta \}\\ ={\left\{L_{*}\in W|\left[\Delta x(t_{l},t_{j}\right)/\sigma_{x}\right]}^{2}+{\left[\Delta y(t_{l},t_{j})/\sigma_{y}\right]}^{2}\leq 2(t_{j}-t_{l})K(t_{l})\} \end{array}$$
where W is defined as the 2D workspace in which the robot searches for the chemical source, η is the probability-density threshold and
(7)$$K(t_{l})=-\ln[2\pi \eta (t_{j}-t_{l})\sigma_{x}\sigma_{y}]$$

The center of the area $OS(t_{l},t_{j})$, denoted by
(8)$$L_{\max}(t_{l})={\left[x_{j}-s_{x}(t_{l},t_{j}),y_{j}-s_{y}(t_{l},t_{j})\right]}^{T}$$
has the maximal probability density in $OS(t_{l},t_{j})$, and the sequence ${\left\{L_{\max}(t_{l})\right\}}_{l=f}^{j}$ forms the most likely C-PP.

### 2.2. Search-Route Planning

The search route is planned online based on the estimated C-PP. Theoretically, the chemical source might be somewhere in the area described by the estimated C-PP. As a rule of thumb, the chemical is more likely to be detected in a downwind location of the source than in an upwind location. Therefore, the search route is designed near and within the downwind area of the estimated C-PP, to be followed by the robot to verify the area covered by the estimated C-PP. Searching by following the online planned route is carried out in the expectation of catching the chemical plume again, to perform the next iteration of searching, guiding the robot to gradually approach the source.

#### 2.2.1. Illustration of the Online Planned Search Route

A schematic diagram of the search-route planning step is shown in Figure 1. The gray strip originating from the chemical source indicates the plume, which is usually unknown to the robot; the ensemble of the blue ellipses denotes the estimated C-PP, and the red dash-dot line represents the most likely C-PP expressed in Equation (8).

The blue-black line $SL_{u}$ and the cyan line $SL_{d}$ constitute the online planned route SL for the robot to follow, i.e., $SL=\{SL_{u},SL_{d}\}$, with the priority decreasing from left to right. $SL_{u}$ and $SL_{d}$, the two parts of the online planned route deviating from the estimated C-PP in the downwind direction, are in roughly the upwind and downwind directions of the robot, respectively. The search route is planned online, aiming to make the robot re-meet the chemical clue. Normally, most areas covered by the estimated C-PP are in the roughly upwind direction of the robot. To be more likely to find chemical clues again, the robot should check these areas first. Therefore, $SL_{u}$ is chosen as the first subroute for the robot to follow.

If another chemical patch is detected during the search along the planned subroutes $SL_{u}$ and $SL_{d}$, a new C-PP will be estimated, and a new search route (including new subroutes $SL_{u}$ and $SL_{d}$) will be planned online for the robot to follow. If there is no chemical patch detected during the subroute $SL_{u}$, the robot has to return to the subroute $SL_{d}$ to check the remaining areas covered by the estimated C-PP. Real-robot experiments show that the robot does not often follow the subroute $SL_{d}$, but it is necessary, because the subroute $SL_{d}$ can be used to deal with two possible problems. One problem is that the robot might search for the chemical source in the wrong direction because of the improper estimation of the C-PP, such as in cases where the assumption of approximately uniform airflow occasionally fails. With the subroute $SL_{d}$, the robot can return to the search area where the most recent chemical-detection event occurred and try for the next possible meeting with the plume. The other problem occurs when the chemical source is near the location of the most recent chemical-detection event, but the robot passes the source, being unaware of its arrival at the source.

Through an iterative searching process (details see Section 2.3), the robot could gradually approach the chemical source. It is worth noting that, if the airflow direction remains stable before and after a new chemical-detection event, the online planned route would be almost a straight line along the upwind direction, and in such a case, the robot would perform a searching behavior just like for a simple upwind action.

#### 2.2.2. Mathematical Description of the Planned Search Route

As mentioned above, according to the current location of the robot and the current wind direction, the online planned route can be divided into two parts, i.e., the subroutes $SL_{u}$ and $SL_{d}$, which are located in the roughly upwind and downwind directions of the robot, respectively. Suppose the robot has detected a chemical patch at $L_{R}(t_{j})$ at time $t_{j}$, with the current time being $t_{k}$ ($t_{j}\leq t_{k}$). $SL_{u}$ and $SL_{d}$ are expressed as
(9-a)$$SL_{u}={\left\{L_{off}(t_{l})||\varphi -\bar{\theta }(t_{k})|>\pi /2\right\}}_{l=j}^{f}$$
(9-b)$$SL_{d}={\left\{L_{off}(t_{l})||\varphi -\bar{\theta }(t_{k})|\leq \pi /2\right\}}_{l=f}^{j}$$
where $L_{off}(t_{l})$ is a point deviating from the estimated C-PP in the downwind direction, φ is the angle of the vector from $L_{R}(t_{k})$ to $L_{off}(t_{l})$, and $\bar{\theta }(t_{k})$ is the short-time-average airflow direction at the current time $t_{k}$. $L_{off}(t_{l})$ is shown in Figure 2 and determined by
(10)$$L_{off}(t_{l})=L_{\max}(t_{l})+\left(d_{ell}(t_{l})+d_{bas}\right)\cdot \left[\begin{array}{c} \cos\bar{\theta }(t_{k})\\ \sin\bar{\theta }(t_{k}) \end{array}\right]$$
where $\left(d_{ell}(t_{l})+d_{bas}\right)\cdot {\left[\cos\bar{\theta }(t_{k}),\sin\bar{\theta }(t_{k})\right]}^{T}$ is the offset from $L_{\max}(t_{l})$ (see Equation (8)), and $d_{bas}$ is an extra offset (set to 0.1 m in our experiments). Here,
(11-a)$$d_{ell}(t_{l})=\sqrt{2(t_{j}-t_{l})K(t_{l})}\cdot \sqrt{{\left(\sigma_{x}\cos\psi \right)}^{2}+{\left(\sigma_{y}\sin\psi \right)}^{2}}$$
(11-b)$$\psi =\tan^{-1}[\sigma_{x}/\sigma_{y}\cdot \tan\bar{\theta }(t_{k})]$$
where ψ is the eccentric angle of the crossing point *X* on the elliptical outline of $OS(t_{l},t_{j})$ in Figure 2, and $\bar{\theta }(t_{k})$ can be determined by
(12)$$\bar{\theta }(t_{k})=\tan^{-1}\left(\sum_{i=f^{\prime }}^{k}u_{y}(L_{R}(t_{i}))/\sum_{i=f^{\prime }}^{k}u_{x}(L_{R}(t_{i}))\right)$$
where ${\left[u_{x}(L_{R}(t_{i})),u_{y}(L_{R}(t_{i}))\right]}^{T}$ is the airflow velocity observed by the robot at position $L_{R}(t_{i})$ at time $t_{i}$, and
(13)$$f^{\prime }=\max(0,k-20)$$

### 2.3. Procedure of CSS

The proposed CSS is an iterative procedure, and its details are shown in Figure 3. The linear search described in Russell et al. [13] is adopted by the robot to find chemical clues, whereby the robot travels at an angle of 35° with respect to the upwind direction. To make the robot move smoothly in the chemical-clue-finding phase, we use the current short-time-average airflow direction as the reference of wind direction. When the robot moves near the boundaries of the search area or encounters an obstacle, it will turn back and make another linear search. When a chemical-detection event occurs, the robot switches to the ERP method to trace the two parts of the newly planned route $SL=\{SL_{u},SL_{d}\}$ with a left-to-right priority. It is worth noting that the search route SL will be updated at each time step according to the short-time-average airflow direction determined by Equation (12), so the route SL tracked by the robot is time-varying. 

During the tracking of the search route SL, whenever a new chemical-detection event occurs, a new C-PP is estimated, and the search route is re-planned. If there is no chemical-detection event after the robot has finished the search-route tracking, the chemical is considered to have been lost, and the robot will switch to the chemical-clue-refinding phase, which uses the same method with the chemical-clue-finding phase. When the distance between the robot and the chemical source is less than 2.5 m (an obstacle-avoidance request would arise in this case), it is considered that “the robot has arrived at the chemical source”, and the robot will be stopped manually. Here, the final stage of CSS, the chemical-source declaration, is left for future work and is not discussed in this paper.

## 3. Experimental Platform

The experimental platform is shown in Figure 4. The robot used was a refitted Pioneer 3-AT (Adept Mobilerobots, Amherst, NH, USA) named MrSOS (Mobile Robot for Searching Odor Source), which is driven and steered differentially by four wheels, two mounted on the left and two on the right. A gas sensor (MiCS 5135, SGX sensor Technology, Co. Ltd., Chelmsford, Essex, UK), an anemometer (Windsonic, Gill Instruments Ltd., Lymington, Hampshire, UK), a laser rangefinder (LMS110, Sick AG, Waldkirch, Baden-Württemberg, Germany), an electronic compass, and a differential GPS (D-GPS) (NovAtel Inc., Calgary, Alta, Canada) module were mounted on the robot. The data from these sensors were received and processed by an onboard computer. The laser rangefinder was mounted at a height of 0.29 m above the ground. The electronic compass and the D-GPS module were used for the mobile-robot localization. Please note that the robot could measure its own speed, and the airflow speed described in this paper was a modified value, with the robot speed being deducted from the anemometer measurement.

The chemical source was a DIY humidifier containing liquid ethanol. The chemical patches were released from an outlet on the top of the humidifier while the liquid ethanol was being atomized by eight ultrasonic units at the bottom of the humidifier. The consumption rate of liquid ethanol was about 47.07 mL/min. 

A binary concentration with an adaptive threshold [30] was used in the experiments to enable the robot to respond to chemical interception quickly and reliably. If the measured concentration is above the adaptive threshold, it indicates a detection event; otherwise, a non-detection event.

## 4. Experiments I—Approximate Uniformity of Wind Field and Probability-Density Threshold 

### 4.1. Testing the Approximate Uniformity of the Local Wind Field

#### 4.1.1. Evaluation Criteria

To verify the approximate uniformity of wind field, a test was performed in an open area near the football field of Tianjin University with two anemometers (Windsonic, Gill Instruments Ltd.) in our earlier work [30]. The two anemometers were mounted in the same horizontal plane, which was 0.6 m above the ground. There were no obstacles around the anemometers. The distance between the two anemometers varied form 1 m to 6 m with an interval of 1 m. For each distance, the wind data were collected for 10 minutes with a sampling frequency of 2 Hz. The maximal and minimal wind speeds we measured were 5.19 m/s and 0.01 m/s, respectively [**Note**: Although the minimal wind speed was 0.01 m/s, actually almost all the data were bigger than 0.2 m/s].

As a measurement of linear correlation between the data collected with the anemometers, the Pearson coefficient *r* for the wind speed and, in the case of the wind direction, the circular-circular correlation index $\rho_{cc}$ for directional data as suggested in Jammalamadaka et al., [31] were computed in this research, instead of the criterion proposed in [30]. Both indexes are bounded between −1 and 1. Correlations equal to 1 or −1, ignoring the sign, correspond to two perfectly correlated variables. A value of zero, on the other hand, implies an absence of a correlation and no relationship between the two variables exists.

The Pearson coefficient *r* is expressed as
(14)$$r=\frac{\sum_{i=1}^{n}\left(v_{1i}-{\bar{v}}_{1}\right)\left(v_{2i}-{\bar{v}}_{2}\right)}{\sqrt{\sum_{i=1}^{n}{\left(v_{1i}-{\bar{v}}_{1}\right)}^{2}}\sqrt{\sum_{i=1}^{n}{\left(v_{2i}-{\bar{v}}_{2}\right)}^{2}}}$$
where $v_{1i}$ and $v_{2i}$ are the *i*-th wind speed sampled from the anemometers 1 and 2, respectively. ${\bar{v}}_{1}$ and ${\bar{v}}_{2}$ stand for the mean of wind speed measured by the anemometers 1 and 2, respectively. 

The circular-circular correlation index $\rho_{cc}$ is expressed as
(15)$$\rho_{cc}=\frac{\sum_{i=1}^{n}\sin\left(\theta_{1i}-{\bar{\theta }}_{1}\right)\sin\left(\theta_{2i}-{\bar{\theta }}_{2}\right)}{\sqrt{\sum_{i=1}^{n}\sin^{2}\left(\theta_{1i}-{\bar{\theta }}_{1}\right)\sin^{2}\left(\theta_{2i}-{\bar{\theta }}_{2}\right)}}$$
where $\theta_{1i}$ and $\theta_{2i}$ are the *i*-th wind direction sampled from the anemometers 1 and 2, respectively. ${\bar{\theta }}_{1}$ and ${\bar{\theta }}_{2}$ stand for the mean of wind directions measured by the anemometers 1 and 2, respectively.

#### 4.1.2. Test Results

From Table 1, we can see that, on the whole, the values of Pearson coefficient and circular-circular correlation index decrease with an increase in the distance between two anemometers. If we classify the wind data according to magnitudes, we can also find that the larger wind speeds lead to stronger correlation. This result is in line with the one obtained in [30]. Most values in Table 1 are bigger than 0.5, so we can say that the wind data of two anemometers are obviously correlative. In other words, the assumption of approximate uniformity of wind field holds in local open areas with a range in diameter of about 5 m in this experiment.

### 4.2. Determining the Probability-Density Threshold η

#### 4.2.1. Evaluation Criterion

To obtain a suitable threshold η in Equation (6), an experiment was designed and conducted in a square where there was no obstacle except the robot and the odor source. The robot was placed in a location in the downwind direction of the odor source to easily have odor detections. During the experiment, the robot did not move and maintained a given distance from the chemical source. The robot continuously measured the airflow velocity and chemical concentration. Let $N_{D}$ represent the number of chemical-detection events. Using a tentative threshold η, within all the $N_{D}$ events, the number of events that the estimated C-PPs contain the source is denoted as $n_{D}$. Thus, the likelihood of the estimated C-PP containing the chemical source can be approximated by the ratio $n_{D}/N_{D}$. In this experiment, $N_{D}$ was set to 100 for each distance from the chemical source. Each chemical-detection event corresponded to an estimated C-PP, i.e., 100 C-PPs were estimated for each distance with a tentative threshold η.

#### 4.2.2. Results and Analysis

The experimental results for different thresholds η and different distances from the source are shown in Figure 5. It is found that the value of $n_{D}/N_{D}$ increases with a decrease in the distance. In addition, $n_{D}/N_{D}$ decreases rapidly when the threshold η is above 10^−2^ m^−2^. The probability-density threshold η influences both the success rate and approaching effectiveness. Using a smaller threshold η for the C-PP estimation might increase the success rate, but it would decrease the approaching effectiveness because the area to be checked will become larger. As a balance between approaching effectiveness and success rate, the threshold η was chosen to be 10^−2^ m^−2^ in this study.

Figure 6 illustrates two of these estimated C-PPs when the distance between the robot and the source was 4 m and the threshold η was $10^{-2}$ m^−2^. In Figure 6, the ellipses mark the boundaries of areas $OS(t_{l},t_{j})$ in Equations (5) and (6). It is found that when the variance of airflow velocity decreases (i.e., the airflow is more stable, see Figure 6a), the estimated C-PP becomes more concentrated in its spatial distribution and the most likely C-PP becomes straighter.

## 5. Experiments II—Chemical Source Searching Based on Online Planned Routes

### 5.1. Experimental Envirofigurenments

The CSS experiments were conducted in two different outdoor field environments, the first one was the football field (limited to 100 m × 60 m) of Tianjin University (named F group), the second one was a small square (limited to 35 m × 55 m) on the north of the Teaching Building No. 26-D&C (named O group). There were no obstacles apart from the chemical source within the search areas for the F group, but for the O group, there was a small parterre (about 9 m × 9 m) standing about 0.4 m above the ground. Figure 7 shows the plan sketch of the second experimental scene. 

### 5.2. Evaluation Criteria

Two criteria were used to evaluate the performance of the proposed CSS strategy. One was named *approaching effectiveness*, and the other is *success rate*. Approaching effectiveness, calculated offline after a trial, is defined as the ratio of the distance approached finally to the real source and the length of the trajectory traveled by the robot in a CSS process. The closer to the real source and the shorter the trajectory travelled by the robot, the higher the approaching effectiveness. The success rate is defined as the ratio of the number of successful trials and the total number of trials.

Figure 8 illustrates how the approaching effectiveness is defined, where the red point *S* represents the chemical source, the point *A* is the initial location of the robot, *B* indicates the location where the first chemical patch is detected, *C* stands for the end location of the robot in the trial. The robot starts with the chemical-clue-finding behavior. Whenever a chemical-detection event occurs, the robot switches to the chemical source tracing phase.

The approaching effectiveness is defined as
(16)$$e_{s}=\left(d_{BS}-d_{CS}\right)/s_{BC}$$
where $d_{BS}$ and $d_{CS}$ denote the linear distances from the position S to B and C, respectively, and $s_{BC}$ represents the trajectory length (curve length) of the robot from B to C. Considering the requirement of obstacle avoidance, $d_{CS}$ cannot be zero. That means the approaching effectiveness is always smaller than 1. For all the CSS experiments, if the approaching effectiveness of a trial is less than 0.2, then the trial is regarded as failed. 

### 5.3. Trials

Thirty-seven trials were conducted in this research, of which 21 trials were in the F group (in the football field), and 16 trials were in the O group (in the field with obstacles). The maximum speed of the robot was set to 1.0 m/s and 0.5 m/s for F and O groups, respectively. Here we want to point out that wind field never has the same spatial distribution twice in natural environments. Therefore, the locations of the first chemical-detection event are different and unknown in advance, even though the robot starts from the same position in these trials. In other words, the start points of using the proposed ERP method for all the trials are different.

#### 5.3.1. Trials in the Football Field (F Group)

Figure 9 shows four scenes corresponding to four different times in one trial of this group, where the chemical source was located at *S* (49.70 m, 29.30 m). During this CSS process, the wind speed ranged from 0.69 m/s to 4.13 m/s, and the wind direction was from 107° to 260°. The robot started from point *A* (4.08 m, 54.10 m) at time 0 s, aiming to find the chemical clue. It met the first chemical patch at point *B* (18.13 m, 17.67 m) at time 56.0 s. The area covered by the ellipses in Figure 9 represents the estimated C-PP, where the red line stands for the most likely C-PP. The planned search route has two parts—$SL_{u}$ and $SL_{d}$—which are represented by the blue-black line and cyan line, respectively. The robot followed subroute $SL_{u}$ and $SL_{d}$ with a left-to-right priority, checking the area covered by the estimated C-PP (see Figure 9a,b). During the tracing phases, new chemical-patches were detected, and then new C-PPs were estimated. At each time step, the search route was re-planned according to the estimated C-PP and the current short-time-average airflow direction.

It is worth noting that in Figure 9b, the wind direction has changed about 40° since the most recent chemical detection event happened. Consequently, the search route was re-planned to make the robot check the area covered by the estimated C-PP as much as possible. When the robot arrived at point *C* (40.74 m, 29.06 m) at time 149.5 s (see Figure 9c), the robot met another chemical patch. Therefore, a new C-PP was estimated, and a new search route was planned and followed by the robot. Finally, the robot approached the chemical source at point *D* (47.40 m, 28.88 m) at time 194.0 s (see Figure 9d). The total tracing time (excluding the chemical-clue-finding time) was 138.0 s, and the approaching effectiveness was 0.87. The video of this trial can be found via the link https://www.youtube.com/watch?v=d69coQvK4ag.

It is found that, with the changes of airflow direction, the robot performed adaptive tracing behavior, checking the possible area where the source might exist according to the current short-time-average airflow direction in expectation of meeting more chemical patches and generating iterative ERP phases. Whenever a new chemical-detection event occurred, the robot re-estimated the C-PP of the newly detected chemical patch, re-planned the search route, and followed it.

The detailed instantaneous wind speeds/directions and chemical detection events are shown in Figure 10, where the circular variance of the wind direction was 0.11. The experimental results for this group of 21 trials are illustrated in Figure 11, where the subgraphs (a) and (b) show the detailed information regarding approaching effectiveness and time cost for each trial, respectively. Overall, in this group, all trials were successful, giving a success rate of 100%. In addition, if assuming the time cost was a linear function of the distance $d_{BS}$ between the starting point *B* to the source location *S* (i.e., $d_{BS}$), ignoring the final distance $d_{CS}$ which is less than 2.5 m, it was approximately 6.88 times of $d_{BS}$. 

#### 5.3.2. Trials in the Field with Obstacles (O Group)

This group of 16 trials was conducted on the north of Teaching Building No. 26-D&C of Tianjin University. The topography of the experimental field is shown in Figure 5, in which a 9m×9m×0.4m (length × width × height) flower bed is in the middle of the square. Real measurements showed that the airflow direction usually changed rapidly and substantially, and the airflow field in the square was frequently far from roughly uniform, and was even more chaotic than the one in the F group due to the impact of the surrounding buildings. In addition, considering the obstacles, such as the flower bed and students who were interested in the robot and stayed in the experimental area, searching the chemical source became more difficult than with the F group. Therefore, during the whole searching process in this group, the local path and global path were planned using the vector field histogram (VFH) [32] method and the search route algorithm proposed in Section 2.2, respectively. 

Figure 12 shows four reconstructed scenes for one trial in the square field. The meanings of lines and marks in Figure 12 are the same as those in Figure 9. The robot started from the point (30.00 m, 8.00 m) at time 0 s, and the source was located at (19.5 m, 22.8 m). The robot moved from east to west (see Figure 5). The first chemical patch was detected at the moment of 62.0 s when the robot moved to the point (28.81 m, 15.41 m). At the moment of 196.0 s, the robot stopped at the point (21.01 m, 22.12 m), which was 1.65 m away from the real source. During the tracing process, the wind speed ranged from 0.01 m/s to 1.88 m/s, and the wind direction −188° to 148°. The total tracing time (excluding the chemical-clue-finding time) was 134 s, and the approaching effectiveness was 0.34. The video of this trial can be found via the link https://www.youtube.com/watch?v=oBlB_b2HqcU.

Lots of gray noise dots (like obstacles) can be found in Figure 12, this is due to at least two reasons. The first is the poor self-localization performance of the robot. Owing to the surrounding high buildings, the D-GPS device could not be used in this trial group. The dead-reckoning method resulted in poor self-localization results. The second is the contribution of students and pedestrians. During the trial process, students who were interested in the robot often stopped in the field. In addition, it was quite normal for pedestrians to pass by the field. 

The detailed instantaneous speeds/directions of airflow and detection events of the chemical patch are shown in Figure 13, where the circular variance of the wind direction was 0.42, from which we can see that the wind direction changed much more frequently than that in the football field. The other obvious feature is that the wind speed was much lower than that for the F group. As mentioned above, this group of trials was conducted in a field surrounded by high buildings, and normally the wind speed in such areas is not very strong, and the wind direction tends to be capricious. The small wind magnitude and fast-changing wind direction resulted in worse uniformity of wind field, leading to lower searching performance. 

The experimental results for this trial group are illustrated in Figure 14, where the subgraphs (a) and (b) show the detailed information regarding approaching effectiveness and time cost for each trial, respectively. Overall, the success rate was 12/16 = 75.0%. The four failed trials’ approaching effectiveness was lower than the given threshold 0.2, which was due to two main reasons. One was the weak wind speed (less than 0.2 m/s) and capricious wind direction, the other was the large cumulative error of the robot’s self-localization using dead reckoning; the D-GPS module was not used, because it almost did not work in this environment, surrounded by high-rise buildings. The time cost was approximately 10.21 times that of $d_{BS}$, with an assumption that the time cost was a linear function of $d_{BS}$ (ignoring the final distance $d_{CS}$). 

#### 5.3.3. Statistical Results and Discussion of the Two Trial Groups

The statistical results of the CSS experiments in two different outdoor environments are listed in Table 2. We can find from Table 2 that, for the group without obstacles (F group), the mean value of approaching effectiveness decreases slightly (except the 1^st^ subgroup, i.e., $d_{BS}$ ranges between 0.0 m and 30.0 m) with an increase in the distance $d_{BS}$. For the group with obstacles (O group), the distances $d_{BS}$ of all trials ranged from 0.0 m to 30.0 m. Compared with group without obstacles (F group), it can be found that in an average sense, the approaching effectiveness decreases observably. The main reason for this is that the robot has to avoid the obstacles during the ERP process, and the trajectory of the robot would be more meandering and longer than the cases without obstacles, thus the approaching effectiveness became lower.

The trials in the O group needed more time per unit searching distance than those in the F group. This can be seen from Figure 11b and Figure 14b, i.e., the time cost per meter in $d_{BS}$ were approximately 6.88 s/m and 10.21 s/m for F and O groups, respectively. This result can be explained from two aspects. Firstly, the maximum speed of the robot was set lower in the O group (0.5 m/s) than in the F group (1.0 m/s). Although the time cost for O group was less than twice the time cost for F group, this does not mean that the search was more efficient in the O group. This is because, especially for the F group, the robot was far from achieving the maximum speed while tracking the dynamic planned search route in the ERP phases due to the high inertia of the robot itself. Secondly, the assumption of approximate uniform airflow field frequently did not hold in most areas of the O group. In addition, the obstacle-avoidance task in the O group resulted in the longest time cost per unit distance.

## 6. Conclusions

Real robot experiments in two different outdoor field environments show that searching for a chemical source according to online planned routes based on estimated C-PPs is a feasible method for a normal wheeled mobile robot moving slower than airflow/chemical plume and is equipped with slow-response gas sensors. In general, searching for a chemical source according to the proposed ERP method shows good robustness to experimental environments. This is because the current and historical flow information are both fully exploited in the proposed method, which results in highly purposive and adaptive search behavior. By following an online planned search route, the robot can systematically check the area covered by the estimated C-PP that possibly contains the chemical source, gradually approaching the chemical source and without having to consider the speed of the chemical plume. Compared with methods trying to make the robot trace chemical plume to the source by keeping contact with the plume, the proposed ERP method has less requirement on the maneuverability of the robot. Because the search route is updated at each time step to match the short-time-average airflow direction, the search behavior of the robot is able to adapt to variation of airflow direction. 

Based on the experimental results, we could further draw the following conclusions. Firstly, the uniformity of airflow field depends, inter alia, on the wind magnitude and distance. Normally, with a decrease in the wind magnitude and an increase in the distance, the uniformity of airflow field decreases. Secondly, uniformity of airflow field has an impact on searching performance. In relatively small and complex fields, since the airflow field is far from uniform, the robot might be confused by the capricious airflow and chemical measurements, leading to a difficult CSS. Thirdly, the proposed ERP method can work in environments with obstacles, although the obstacles decrease the approaching effectiveness and increase the time cost. Lastly, the approaching effectiveness of the proposed method is not sensitive to the maximum speed of the robot. 

## Figures and Tables

**Figure 1 sensors-19-00426-f001:**
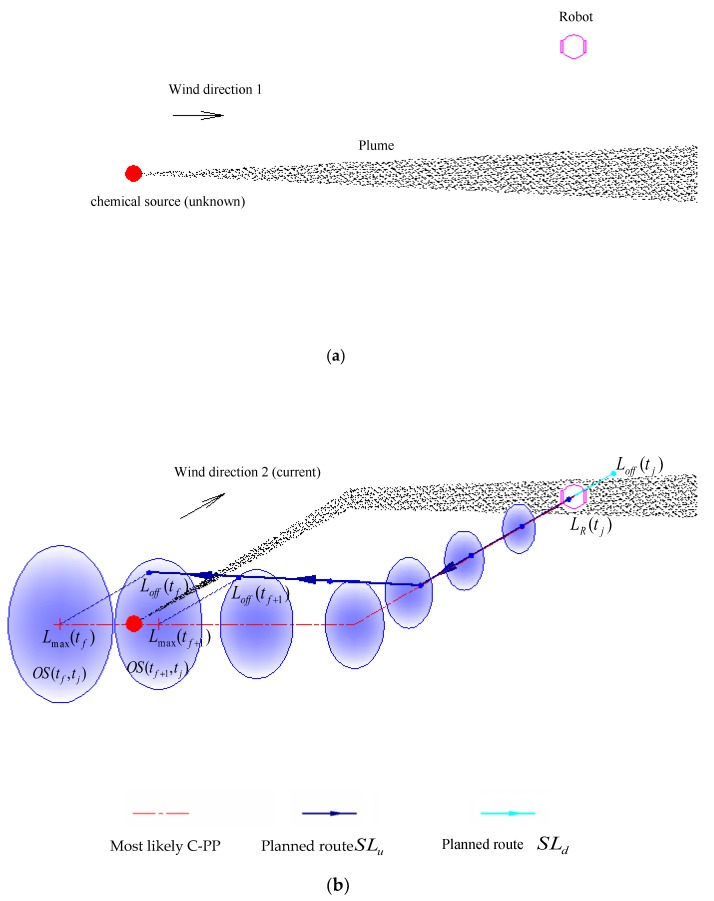
Chemical-source localization by following the online planned route. The planned route is represented by two colors, i.e., blue-black and cyan, corresponding to the two different parts $SL_{u}$ and $SL_{d}$, respectively. The wind initially blows from left to right (marked with “Direction 1”) for a period in (**a**), and then changes by about 30 degrees (marked with “Direction 2”) for another period in (**b**). It is assumed that the wind is uniform throughout the planar space around the robot during each period (See 4.1 for more details). All chemical patches previously released from the chemical source therefore form the plume shown by the gray strip. The chemical patches in (**a**) are collectively transported in wind direction 2 in (**b**). When a chemical-detection event happens in (**b**), a C-PP, i.e., the ensemble of the elliptic areas $\left\{OS(t_{l},t_{j}),l=f,f+1,\ldots ,j-1\right\}$, is estimated, and then a search route is planned online to make the robot attempt to re-meet the chemical clue. The robot will finally approach the source through an iterative searching procedure.

**Figure 2 sensors-19-00426-f002:**
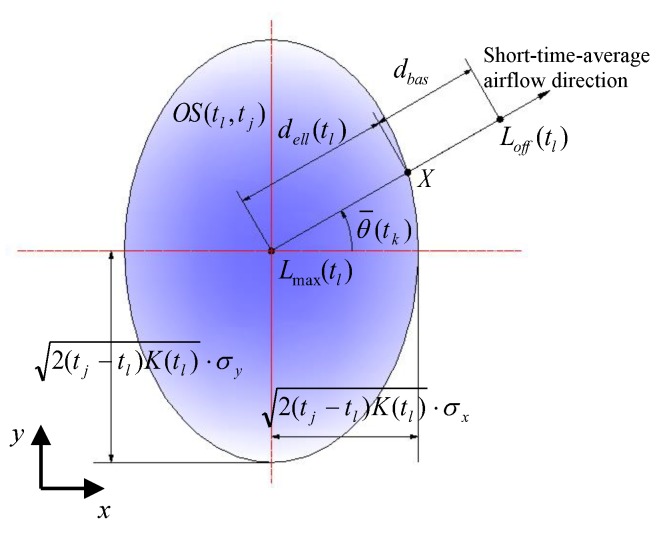
Generation of the deviation-path point $L_{off}(t_{l})$.

**Figure 3 sensors-19-00426-f003:**
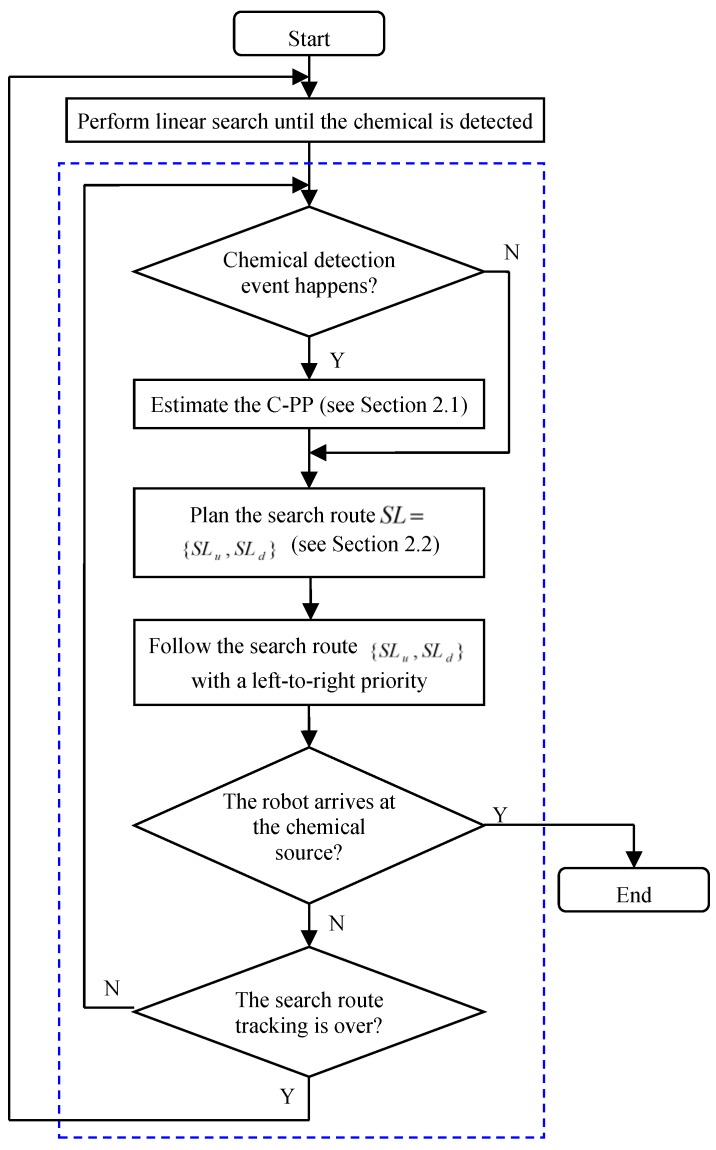
Framework of the CSS based on ERP method. The ERP procedure is represented by the steps enclosed within broken lines.

**Figure 4 sensors-19-00426-f004:**
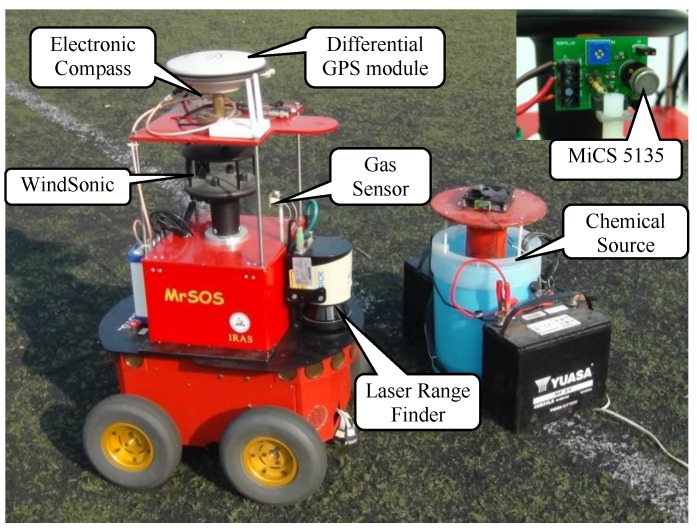
Experimental platform: the wheeled mobile robot MrSOS and the onboard sensors. An enlarged view of the gas sensor (MiCS 5135) is shown at the top right corner.

**Figure 5 sensors-19-00426-f005:**
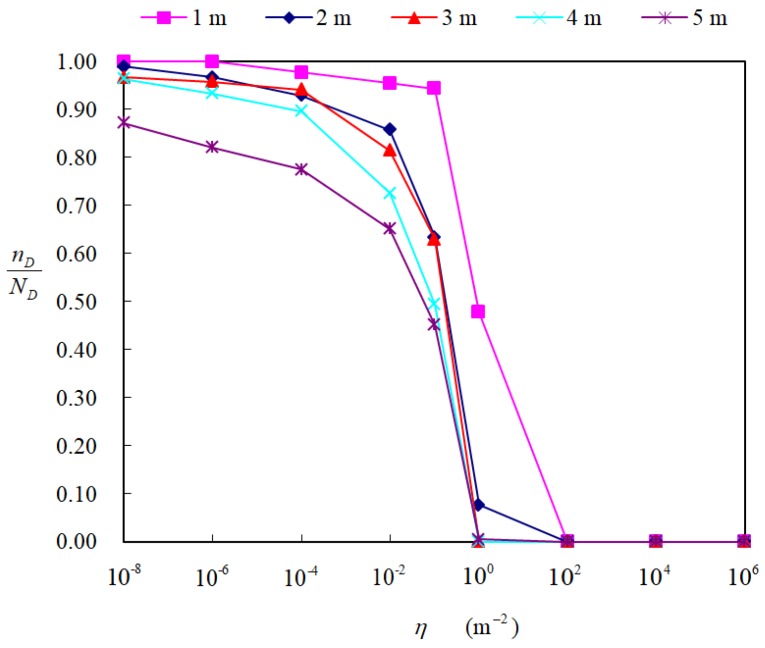
The value of $n_{D}/N_{D}$ for eight different thresholds η and five different distances from the source.

**Figure 6 sensors-19-00426-f006:**
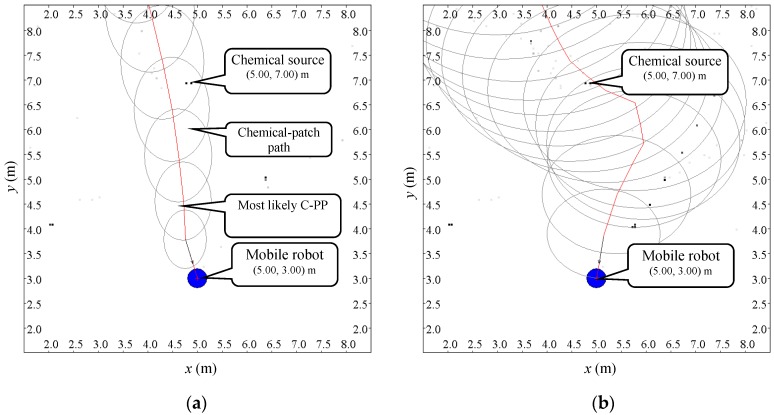
Two of the 100 estimated C-PPs when the distance between the robot and the source was 4 m and the threshold η in Equation (6) was chosen to be $10^{-2}$ m^−2^. The ellipses indicate the boundaries of areas $OS(t_{l},t_{j})$ in Equations (5) and (6). (**a**) A case of the airflow being more stable ($\left[\sigma_{x}^{2},\sigma_{y}^{2}]=[0.029,0.053\right]$ m^2^/s^2^). (**b**) A case of the airflow being less stable ($\left[\sigma_{x}^{2},\sigma_{y}^{2}]=[0.260,0.160\right]$ m^2^/s^2^).

**Figure 7 sensors-19-00426-f007:**
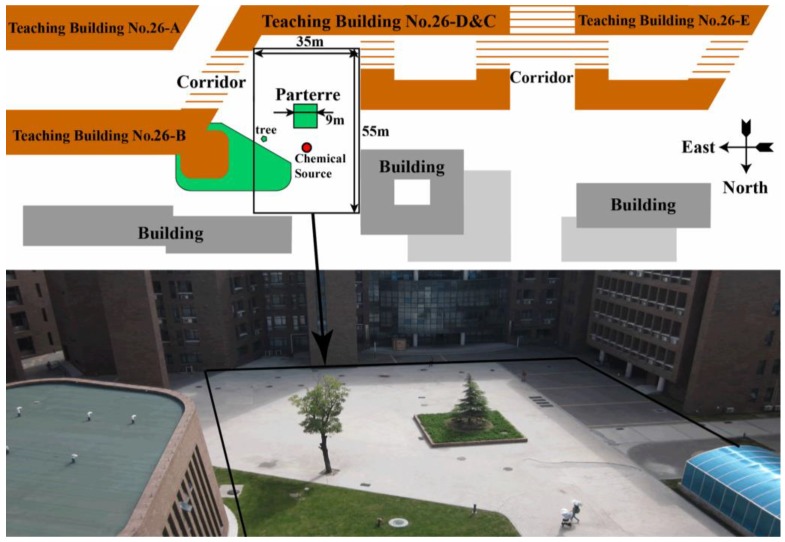
Plan sketch (upper) and picture (lower) of the experimental field for the O group. The trials of the O group were conducted in the small square (limited to 35 m × 55 m) on the north of Teaching Building No. 26-D&C, and there is a small garden standing about 0.4 m above the ground.

**Figure 8 sensors-19-00426-f008:**
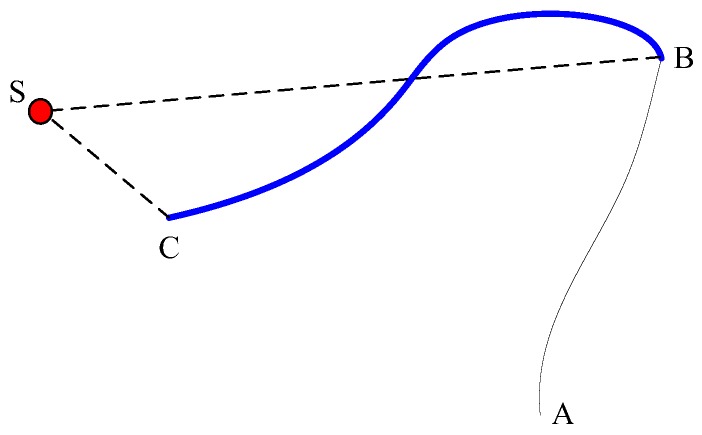
Sketch of the approaching effectiveness definition. The trajectory from *A* to *B* indicates the chemical-clue-finding phase, and *B* to C belongs to the plume tracing / traversal phase. The red point *S* stands for the chemical source. For our research, the plume tracing / traversal phase corresponds to the odor source searching with the ERP method and might include the chemical-clue-refinding phase.

**Figure 9 sensors-19-00426-f009:**
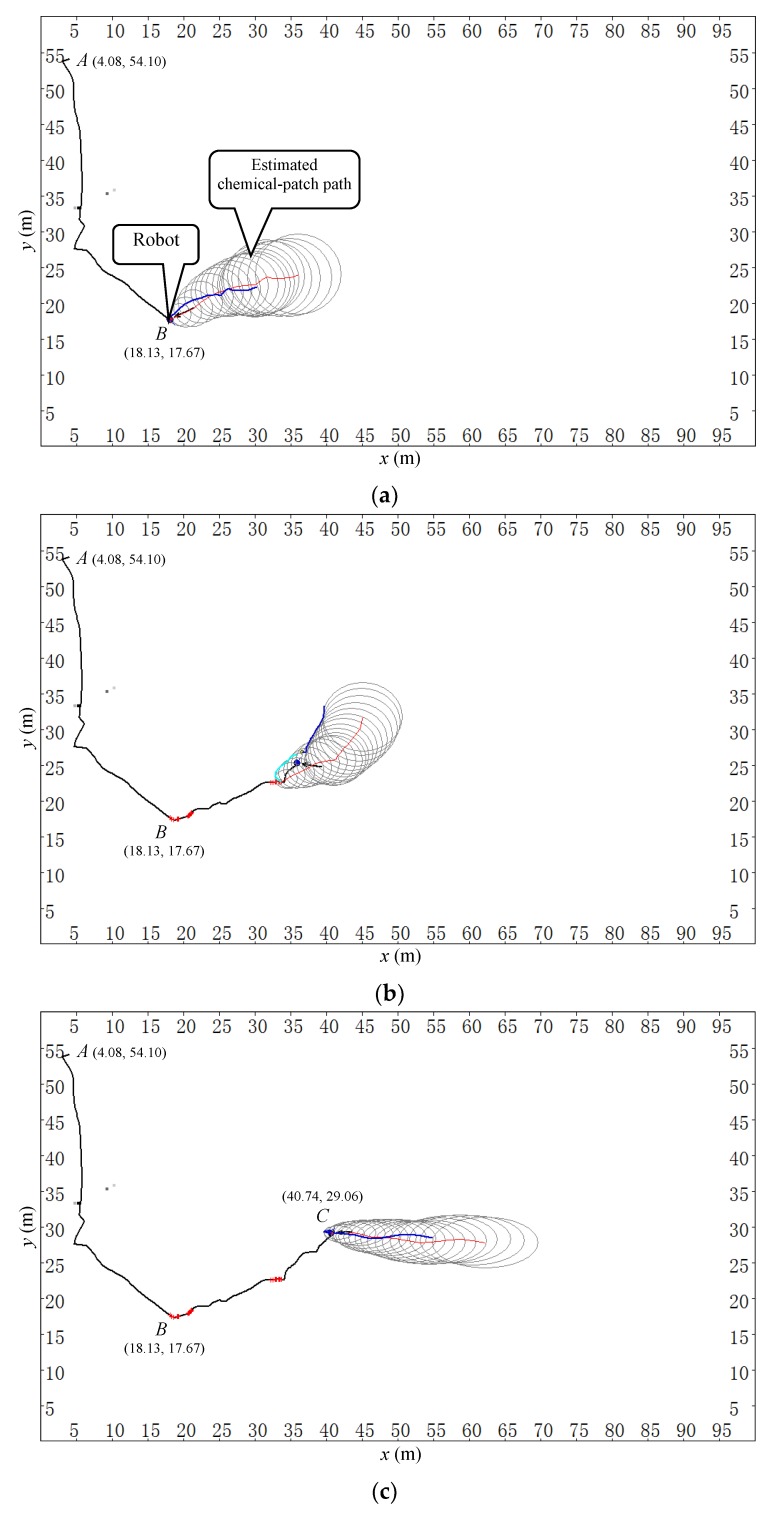
The process of chemical-clue finding and ERP procedure in the football field of Tianjin University. (**a**) *t* = 56.0 s. (**b**) *t* = 125.5 s. (**c**) *t* = 149.5 s. (**d**) *t* = 194.0 s. ‘*A*’ denotes the start position of the robot, ‘*S*’ indicates the true location of the chemical source, and ‘*B*’ is where the ERP procedure was launched. The planned route is represented by two colors, i.e., blue-black and cyan, corresponding to the two different parts $SL_{u}$ and $SL_{d}$, respectively. Along the trajectory of the robot, each symbol ‘+’ represents a chemical-detection event. The solid arrow near the robot denotes the instantaneous wind direction observed by the robot. The trajectory of the robot from points ‘*A*’ to ‘*B*’ belongs to the chemical-clue-finding phase and the remaining part belongs to the ERP procedure. The timer began when the robot started to find chemical clue at point ‘*A*’. The total tracing time (excluding the chemical-clue-finding time) was 138.0 s.

**Figure 10 sensors-19-00426-f010:**
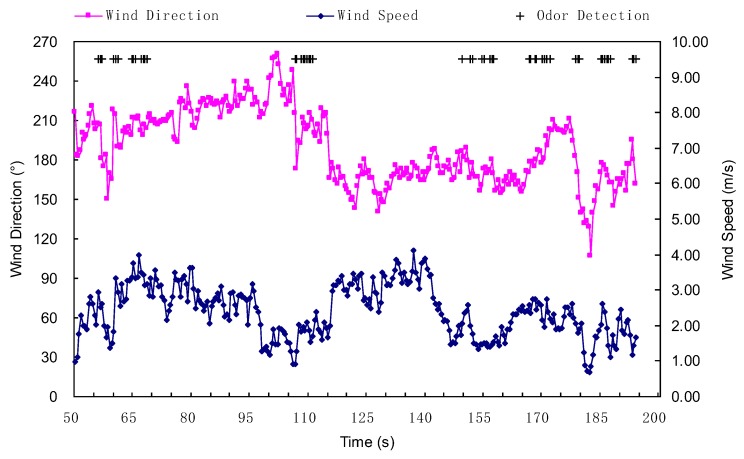
Instantaneous wind directions/speeds and chemical-detection events during the ERP phase shown in Figure 9.

**Figure 11 sensors-19-00426-f011:**
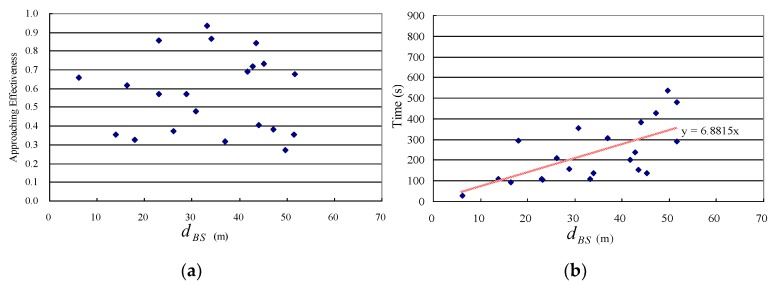
Results of 21 ERP experiments in the F group. (**a**) Approaching effectiveness. (**b**) Time costs. Each mark in the two subgraphs stands for the corresponding value of an experiment.

**Figure 12 sensors-19-00426-f012:**
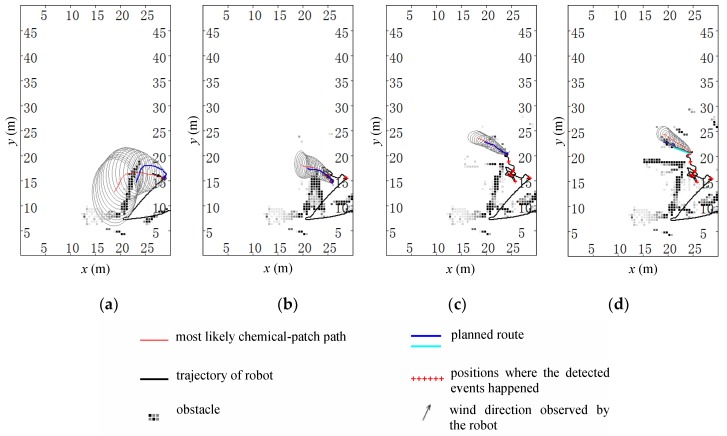
Four reconstructed scenes during the CSS process in the square field. (**a**) *t* = 62.0s, the robot was at (28.81, 15.41) m. (**b**) *t* = 132.5s, the robot was at (25.80, 15.07) m. (**c**) *t* = 173.0s, the robot was at (24.06, 20.28) m. (**d**) *t* = 196.0s, the robot was at (21.01, 22.12) m. The unit of each subgraph is meters. The total tracing time (excluding the chemical-clue-finding time) was 134.0 s.

**Figure 13 sensors-19-00426-f013:**
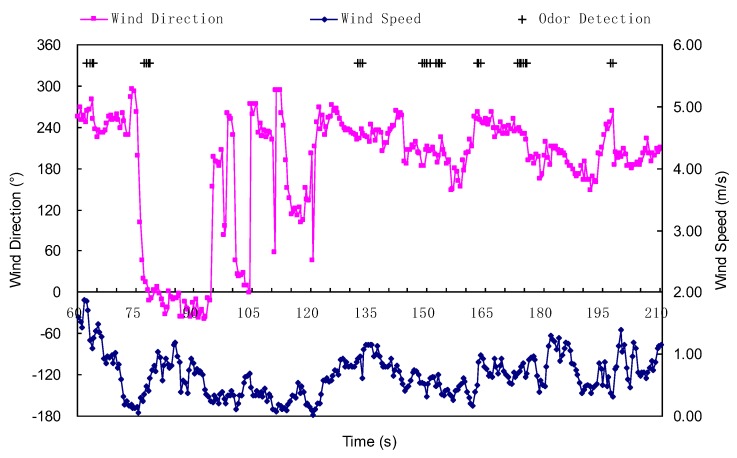
Instantaneous wind directions/speeds and chemical-detection events during the ERP phase shown in Figure 12.

**Figure 14 sensors-19-00426-f014:**
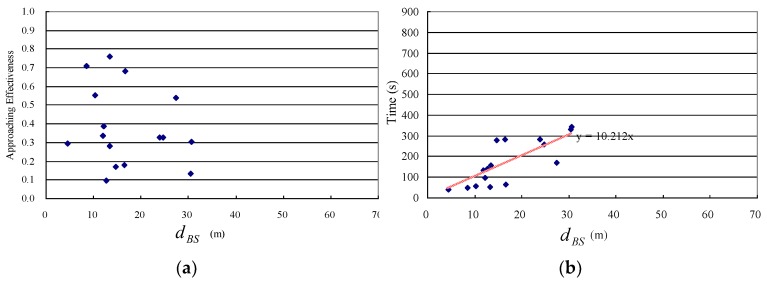
Results of 16 ERP experiments in the O group. (**a**) Approaching effectiveness. (**b**) Time cost. Each mark in the two subgraphs stands for the corresponding value of an experiment.

**Table 1 sensors-19-00426-t001:** Pearson coefficient of wind speeds and circular-circular correlation index of wind directions.

*d* (m)	Pearson Coefficient	Circular-Circular Correlation Index
1	0.9556	0.9048
2	0.8312	0.9091
3	0.6505	0.5642
4	0.4909	0.5673
5	0.5604	0.6396
6	0.5475	0.2106

*d*: distance between two anemometers.

**Table 2 sensors-19-00426-t002:** Statistical results for the ERP method in outdoor environments.

Groups	F Group (Trials in the Football Field)				O Group (Trials in the Square)			
	N	$d_{BS}\;(m)$	$eff_{a}$	T (s)	N	$d_{BS}\;(m)$	$eff_{a}$	T (s)
0.0–30.0 (1st subgroup)	8	19.4 ± 6.1	0.54 ± 0.15	138.1 ± 68.4	16	17.0 ± 4.2	0.38 ± 0.11	171.4 ± 57.9
30.0–50.0 (2nd subgroup)	11	40.8 ± 4.2	0.60 ± 0.16	271.3 ± 94.8	0			
50.0–70.0 (3rd subgroup)	*2*	*51.6*	*0.51*	*385.0*	0			

* The statistical results of the trials are given as means with 95% confidence intervals for $d_{BS}$, approaching effectiveness and time cost. “N” means the number of trials in the corresponding subgroup, $eff_{a}$ the approaching effectiveness, and “T” the time cost. The items in italics are given only as means without 95% confidence intervals, which corresponds to the groups that the number of trials is less than 3. The maximum speed of the robot was set to 1.0 m/s and 0.5 m/s for F group and O group, respectively.

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
