# Peer review of "Chemical Source Searching by Controlling a Wheeled Mobile Robot to Follow an Online Planned Route in Outdoor Field Environments†"

_sensors, 2019, doi:10.3390/s19020426_

Reviewer 1 Report

The authors has proposed a new method for gas source localization using a mobile robot. The proposed method is based on chemical patch path estimation, which was originally proposed in the authors' previous paper. However, a different approach is proposed in this paper to plan the search path for the robot from the estimated chemical patch path. The results presented in this paper are impressive, and I believe the paper will be of great interest to the reader of this journal. However, there are several points that remained unclear for me after reading the manuscript. My concerns are listed below.

(1) Title

I think the title is a bit misleading. From the title ("controlling a wheeled mobile robot to follow a planned route"), I thought at first the robot was moved along a path, which had been determined a priori before starting the experiments.

(2) Page 3, Line 108.

The authors are implicitly assuming that the wind field and gas dispersal are all in 2D, but it would be better to clearly state this assumption in the text.

(3) Equations (3-a) and (3-b).

What do "L"s in those equations represent?

(4) Subsections 2.2.1 and 2.2.2.

These two subsection can be merged because there are many duplicated sentences.

(5) Equations (9-a) and (9-b)

Please elaborate on these equations. Do they mean that the set, { Loff(tl) }, is divided into two subsets according to the location of Loff(tl) with respect to the robot position and the wind direction? It had not been able to fully understand those equations until I saw Fig. 9.

(6) Fig. 3.

Please explain how the direction of the robot in "the chemical-clue-finding phase" is determined. I think the choice of the initial detection had a large impact on the results of the experiments.

(7) Page 8, Line 253. "The robot used was a refitted Pioneer 3-DX"

From Fig. 4, I think the robot is not based on Pioneer 3-DX, but on Pioneer 3-AT.

(8) Page 8, Line 266. "The release rate of the chemical source was about 47.07 ml/min."

Please clarify whether this release rate is the volume flow rate of ethanol vapor or the consumption rate of liquid ethanol.

(9) Fig. 4.

Please adjust the size of the box containing the label, "Chemical Source."

(10) Page 9, Line 279. "a small parterre standing about 0.4 m above the ground"

Please also let the readers know the height of the laser scan by the laser range finder.

(11) Section 3.3.

It seems that the condition for a trial to be regarded as failed is not consistent with the description of the ERP procedure in Section 2. On Page 10, Line 306, it is written that the trial was regarded as failed if the final distance between the robot and the chemical source exceeds 2.5 m. However, according to the descriptions in Section 2.3 and Fig. 3, the search is endlessly repeated unless the robot comes within 2.5 m radius from the chemical source. No condition to terminate this loop is presented in this paper. 

Moreover, the definition of the approaching effectiveness appears to assume that the source is found in a single series of gas detection events. How this approaching effectiveness is calculated if the robot switch back to the chemical clue finding phase?

(12) Page 11, Line 339. "the assumption of approximate uniformity of wind field holds in open areas"

In this paper, the authors are discussing about chemical source search in "thousands-of-square-meter outdoor environments," but the correlation of the wind velocity is tested only up to 10 m. Please provide some discussions about this issue.

(13) Section 4.2. "In this experiment, ND was set to 100 for each distance from the chemical source."

Please explain the experimental setup in more detail. Where the experiment was conducted? In which direction from the source the robot was placed?

(14) Fig. 12(a).

According to the main text, Fig. 12(a) shows the position of the robot when the first chemical patch was detected. The robot was supposed to perform linear search until the first chemical patch was detected, but why the trajectory of the robot in Fig. 12(a) is kinked?

(15) Fig. 13

The authors state that the wind direction shown in Fig. 13 is changing much more frequently. However, I suspect this is an artifact. It appears from the graph that the wind direction changed significantly at around 75 s. However, the real variations in the wind direction would be much smaller if the graph is re-made considering the fact that 360 degree and 0 degree represents the same angle. It seems to me that the wind variations in Fig. 13 are slightly smaller than those in Fig. 10. It would be nice if the authors can provide more objective indices on wind variations, e.g., the circular variance.

(16) Page 19, Line 521. "the 95% confidence intervals of approaching effectiveness and time cost generally become wide with an increase in the mean of dBS."

It appears to me that the 95% confidence intervals of dBS and and approaching effectiveness remains same for all ranges of dBS in Table 2. Moreover, the 95% confidence interval of time cost also appears to decrease with dBS: the interval is about 1/2 of the mean for 0-30 m, but is about1/3 for 30-50 m.

I also wonder what the point on discussing the confidence interval for dBS. I guess the variations in dBS largely depends on the choice of initial direction of the robot in the chemical-clue-finding phase.

(17) Page 19, Line 529. "The trials in O group needed more time per unit searching distance than those in F group."

In trials in O group, the maximum speed of the robot was reduced by half. However, the time cost was less than twice of the cost for F group. Doesn't it mean that the search was more efficient in O group? How can it be possible even though the approaching effectiveness was low and the robot spent more time avoiding obstacles in O group?

(18) Reference 30.

It would be nice if the comparison can be made between the results presented in this paper and in reference 30. For example, How did the actual paths of the robot differ and why did the success rate increase?

Author Response

Attached here please find the response letter for you. Thank you very much for your great help.

Reviewer 2 Report

The paper is very good. The experiment is precise and the results are reliable. I had some small questions but in the course of my reading the authors give very good answers

Author Response

Attached here please find the response letter for you. Thank you very much for your help.

Reviewer 3 Report

This paper presents the chemical source searching in outdoor field environments via a wheeled robot. The idea is well presented and paper is written in a prolific manner. I believe the paper would be helpful for the wide readers of sensors, hence, I want to accept its publication after the minor revision of the comments provided in the next:

1)      There are minor errors in the text editing such as line 42 [2, 8]. There should not be a space in between the consecutive references as per MDPI sensors’ formal referencing style. Review for the rest of the paper as well.

2)      It’s better to follow the formal MDPI sensors’ manuscript template to structure the manuscript. The section 3 and 4 are confusing where the results are imbibed into the experimental section. Better to make a single experimental section and explain the results in a separate section for readers’ understanding.

3)      The infamous and non-derived equations need to be cited with a proper bibliographic reference for readers’ understanding.

Author Response

Attached here please find the response letter for you. Thank you very much for your great help.

Round  2

Reviewer 1 Report

I think the paper is almost ready for publication, but unfortunately, the answer to point 11 in my first review report is not satisfactory. I asked the authors to clarify the conditions for terminating the search loop, but I still can get full understanding on this issue after carefully reading the response from the authors. Because an error in defining the conditions for terminating the search would have a large impact on the search success rate, I recommend "reconsider after minor revision" once again rather than "accept after minor revision."

I hope the authors can give me a clear explanation on this issue because the other part of the paper is well written and also the contents of the paper is of great interest to the readers of the journal. My concern is as follows. According to Lines 248-251 in the revised manuscript, the trial is regarded as "failed" and the robot is stopped manually when the approaching effectiveness is less than 0.2. Fig. 3 in the revised manuscript also shows that the search is terminated when the robot arrives at the chemical source or the approaching effectiveness is less than 0.2. However, the approaching effectiveness is defined as (dBS – dCS) / SBC where C is "the end location of the trial" (Lines 371-374). If this explanation is correct, the approaching effectiveness can be calculated ONLY AFTER the trial is terminated (and therefore the end point is determined). How can this value be used for terminating the loop in Fig. 3? The termination of the loop must be judged by the values available online.

 If the approaching effectiveness is calculated online by taking the current location of the robot as C, then its value always becomes less than 0.2 right after the first chemical patch is detected. Therefore, the search would be terminated immediately.

The only reasonable explanation that I can think of is that C actually represents the end location of each plume tracing/traversal phase. This is why I wrote in my first review report that "the definition of the approaching effectiveness appears to assume that the source is found in a single series of gas detection events." If this is the case, however, it means that there are two different kinds of "approaching effectiveness" in this paper: one calculated for each plume tracing/traversal phase and one calculated for overall search trajectory (those presented in Table 2).

If I am wrong, please correct me and provide clear explanation in the revised text.

Author Response

 Attached here please find our revised manuscript.

Round  3

Reviewer 1 Report

Thanks for clarification. I think the paper is now ready for publication.